# Unlocking the Hidden Genetic Diversity of Varicosaviruses, the Neglected Plant Rhabdoviruses

**DOI:** 10.3390/pathogens11101127

**Published:** 2022-09-29

**Authors:** Nicolas Bejerman, Ralf G. Dietzgen, Humberto Debat

**Affiliations:** 1Instituto de Patología Vegetal, Centro de Investigaciones Agropecuarias, Instituto Nacional de Tecnología Agropecuaria (IPAVE—CIAP—INTA), Camino 60 Cuadras Km 5.5, Córdoba X5020ICA, Argentina; 2Consejo Nacional de Investigaciones Científicas y Técnicas, Unidad de Fitopatología y Modelización Agrícola, Camino 60 Cuadras Km 5.5, Córdoba X5020ICA, Argentina; 3Queensland Alliance for Agriculture and Food Innovation, The University of Queensland, St. Lucia, QLD 4072, Australia

**Keywords:** plant rhabdovirus, varicosaviruses, genome architecture, virus taxonomy, metatranscriptomics

## Abstract

The genus *Varicosavirus* is one of six genera of plant-infecting rhabdoviruses. Varicosaviruses have non-enveloped, flexuous, rod-shaped virions and a negative-sense, single-stranded RNA genome. A distinguishing feature of varicosaviruses, which is shared with dichorhaviruses, is a bi-segmented genome. Before 2017, a sole varicosavirus was known and characterized, and then two more varicosaviruses were identified through high-throughput sequencing in 2017 and 2018. More recently, the number of known varicosaviruses has substantially increased in concert with the extensive use of high-throughput sequencing platforms and data mining approaches. The novel varicosaviruses have revealed not only sequence diversity, but also plasticity in terms of genome architecture, including a virus with a tentatively unsegmented genome. Here, we report the discovery of 45 novel varicosavirus genomes which were identified in publicly available metatranscriptomic data. The identification, assembly, and curation of the raw Sequence Read Archive reads has resulted in 39 viral genome sequences with full-length coding regions and 6 with nearly complete coding regions. The highlights of the obtained sequences include eight varicosaviruses with unsegmented genomes, which are linked to a phylogenetic clade associated with gymnosperms. These findings have resulted in the most complete phylogeny of varicosaviruses to date and shed new light on the phylogenetic relationships and evolutionary landscape of this group of plant rhabdoviruses. Thus, the extensive use of sequence data mining for virus discovery has allowed us to unlock of the hidden genetic diversity of varicosaviruses, the largely neglected plant rhabdoviruses.

## 1. Introduction

A recently discovered huge number of diverse viruses has revealed the complexities of the evolutionary landscape of replicating entities and the challenges associated with their classification [1], leading to the first comprehensive proposal of the virus world megataxonomy [2]. Nevertheless, a minuscule portion, likely a small fraction of one percent, of the virosphere has been characterized so far [3]. Therefore, we have a limited knowledge of the vast world virome, with its remarkable diversity, that includes every potential host organism assessed so far [4,5,6]. Data mining of publicly available transcriptome datasets has become an efficient and inexpensive strategy to unlock the diversity of the plant virosphere [5]. Data-driven virus discovery relies on the vast number of available datasets on the Sequence Read Archive (SRA) of the National Center for Biotechnology Information (NCBI). This resource, which is growing at an exceptional rate and includes data of a large and diverse number of organisms, represents a substantial fraction of species that populate our planet, which makes the SRA database an invaluable source to identify novel viruses [7].

*Varicosavirus* is one of the six genera that are comprised of plant rhabdoviruses (family *Rhabdoviridae*, subfamily *Betarhabdovirinae*), and its members are thought to have a negative-sense, single-stranded, bi-segmented RNA genome [8]. Nevertheless, recently, we described the first apparently unsegmented varicosavirus [9]. In those varicosaviruses with segmented genomes, RNA 1 consists of one to two genes, with one of those encoding the RNA-dependent RNA polymerase L, while RNA 2 consists of three to five genes, with the first open reading frame (ORF) encoding a nucleocapsid protein (N) [8,10]. On the other hand, the only unsegmented varicosavirus described so far has five ORFs, in the order: 3′-N-Protein 2-Protein 3-Protein 4-L-5′ [9]. Varicosaviruses appear to have a diverse host range that includes dicots, monocots, gymnosperms, ferns, and liverworts [6,9]. The vector of a sole member, lettuce big vein-associated virus (LBVaV), has been characterized, which is the chytrid fungus *Olpidium* spp. [11].

Until 2017, LBVaV was the only identified and extensively characterized varicosavirus [12,13,14], and then, in 2017 and 2018, two novel varicosaviruses were identified through high-throughput sequencing (HTS) [15,16]. However, in 2021 and 2022, there was a five-fold increase in the number of reported varicosaviruses, with 12 out 15 discovered through data mining of publicly available transcriptome datasets [6,9,17,18], while the other three were identified using HTS [19,20,21] (Appendix A). Nevertheless, only some minor biological aspects, such as mechanical transmissibility, of some of these members were further characterized [15,20]. Therefore, varicosaviruses are, by far, the least-studied plant rhabdoviruses, and many aspects of their epidemiology remain elusive. In terms of genetic diversity, before this study, while greatly expanded by recent works, the *Varicosavirus* genus includes only three accepted species and 15 tentative members.

In this study, we identified 45 novel varicosaviruses by analyzing publicly available metatranscriptomic data. Thus, the extensive use of data mining for virus discovery has allowed us to unlock some of the hidden diversity of varicosaviruses, the much-neglected plant rhabdoviruses.

## 2. Material and Methods

### 2.1. Identification of Plant Rhabdovirus Sequences from Public Plant RNA-seq Datasets

Three strategies were used to detect varicosavirus sequences: (1) Amino acid sequences corresponding to the nucleocapsid and polymerase proteins of known varicosaviruses were used as queries in tBlastn searches with the parameters word size = 6, expected threshold = 10, and scoring matrix = BLOSUM62, against the Viridiplantae (taxid: 33090) Transcriptome Shotgun Assembly (TSA) sequence databases. The obtained hits were manually explored and based on percentage identity, query coverage, and E-value (>1 × 10^−5^) and shortlisted as likely corresponding to novel virus transcripts, which were then further analyzed. (2) Raw sequence data corresponding to the SRA database associated with the 1K study [22] were explored for varicosa-like virus sequences. (3) The Serratus database was explored, employing the serratus explorer tool [5], and using as queries the sequences of LBVaV, red clover varicosavirus, and black grass varicosavirus. Those SRA libraries that matched the query sequences (alignment identity > 45%; score > 10) were further explored in detail.

### 2.2. Sequence Assembly and Identification

The nucleotide (nt) raw sequence reads from each SRA experiment, which are associated with different NCBI bioprojects (Table 1), were downloaded and pre-processed by trimming and filtering with the Trimmomatic tool as implemented in http://www.usadellab.org/cms/?page=trimmomatic (accessed on 19 August 2022). The resulting reads were assembled de novo with rnaSPAdes using standard parameters on the Galaxy.org server. The transcripts obtained from the de novo transcriptome assembly were subjected to bulk local BLASTX searches (E-value < 1 × 10^−5^) against a collection of varicosavirus protein sequences available at https://www.ncbi.nlm.nih.gov/protein?term=txid140295[Organism] (accessed on 19 August 2022). The resulting viral sequence hits of each bioproject were visually explored. Tentative virus-like contigs were curated (extended or confirmed) by iteratively mapping each SRA library’s filtered reads. This strategy used BLAST/nhmmer to extract a subset of reads related to the query contig and used the retrieved reads to extend the contig and then repeated the process iteratively using the extended sequence as query. The extended and polished transcripts were reassembled using the Geneious v8.1.9 (Biomatters Ltd., San Diego, CA, USA) alignment tool with high sensitivity parameters. Bowtie2, available at http://bowtie-bio.sourceforge.net/bowtie2/index.shtml (accessed on 26 September 2022), was used with standard parameters for filtered read mapping to calculate the mean coverage of each assembled virus sequence.

### 2.3. Bioinformatics Tools and Analyses

#### 2.3.1. Sequence Analyses

ORFs were predicted with ORFfinder (minimal ORF length 150 nt, genetic code 1, https://www.ncbi.nlm.nih.gov/orffinder/, accessed on 22 August 2022) and the functional domains and architectures of translated gene products were determined using InterPro (https://www.ebi.ac.uk/interpro/search/sequence-search, accessed on 22 August 2022) and the NCBI conserved domain database-CDD v3.19 (https://www.ncbi.nlm.nih.gov/Structure/cdd/wrpsb.cgi, accessed on 22 August 2022). Further, HHPred and HHBlits, as implemented in https://toolkit.tuebingen.mpg.de/#/tools/ (accessed on 22 August 2022), were used to complement the annotation of divergent predicted proteins by hidden Markov models. Transmembrane domains were predicted using the TMHMM version 2.0 tool (http://www.cbs.dtu.dk/services/TMHMM/, accessed on 22 August 2022).

#### 2.3.2. Pairwise Sequence Identity

Percentage amino acid (aa) sequence identities of the L protein of those varicosaviruses identified in this study, as well as those available in the NCBI database, were calculated using SDTv1.2 [59]. Virus names, abbreviations, and NCBI accession numbers of the varicosaviruses already reported are shown in Appendix A.

#### 2.3.3. Phylogenetic Analysis

Phylogenetic analysis based on the predicted polymerase protein of all available varicosaviruses was completed using MAFFT 7.505 (https://mafft.cbrc.jp/alignment/software) (accessed on 25 August 2022) with multiple aa sequence alignments and using FFT-NS-i as the best-fit model. The aligned aa sequences were used as inputs to generate phylogenetic trees using the maximum-likelihood method (best-fit model = E-INS-i) with the FastTree 2.1.11 tool (available at http://www.microbesonline.org/fasttree/) (accessed on 25 August 2022). Local support values were calculated with the Shimodaira-Hasegawa test (SH) and 1000 trees were resampled. The L proteins of four selected cytorhabdoviruses were used as outgroups. To explore the potential phylogenetic co-divergence of varicosaviruses with their associated host plants, plant host cladograms were generated in phyloT v.2 (https://phylot.biobyte.de/, accessed on 26 August 2022) based on NCBI Taxonomy. Connections were manually inferred between the viral and plant phylograms and cladograms and visually inspected.

## 3. Results and Discussion

Most varicosaviruses likely do not induce easily discernable disease symptoms. Since their presence is not expected in the sequencing libraries of apparently “healthy” vegetables, they are ideal candidates to be identified through mining publicly available metatranscriptomic data. Accordingly, very recently, 12 novel proposed varicosaviruses were discovered when publicly available transcriptome datasets were mined [6,9,17,18]. Therefore, to unlock the hidden diversity of varicosaviruses, we extensively searched for these viruses in already available plant transcriptome data. This bioinformatics research resulted in the identification of 45 novel varicosaviruses, including the corrected full-length coding genome segments of the previously reported Arceuthobium sichuanense-associated virus 2 (ASaV2) [18], which had apparently been reconstructed from the genome segments of two different varicosaviruses. We also identified three novel variants of three recently discovered varicosaviruses, confirming and strengthening the results previously reported by Bejerman et al. [9]. This significant number of newly discovered varicosaviruses represents a 3.5-fold increase in the known varicosaviruses (Appendix A), which clearly highlights the importance of data-driven virus discovery to illuminate the landscape of largely overlooked taxonomic groups, such as varicosaviruses.

More details, identification, assembly, and curation of raw SRA reads in this study resulted in 39 viral genome sequences with full-length coding regions and six with nearly complete coding regions. These viruses were associated with 45 plant host species (Table 1). Most of the tentative plant hosts of the novel varicosaviruses are herbaceous dicots (24/45), nine are herbaceous monocots, eight are gymnosperms, and four are liverworts and ferns (Table 1).

The genomes of 37 viruses identified in this study were bisegmented, where the RNA 1 of 36 of them encodes only the L protein, while the RNA 1 of Chamaemelum virus 1 (ChaV1) has an additional ORF 5’ to the L gene, supported by the identification of the conserved intergenic sequence (see below), encoding a 171 aa putative protein (Table 1, Figure 1), which appears to be the first varicosavirus reported with an ORF in this position. The RNA 2 segments of these 37 viruses have three to five genes in the order 3′-N-PX-5′. Twelve of them have three genes, while 17 have four genes and eight contained five genes (Table 1, Figure 1). Of the previously reported varicosaviruses, six have three genes, four have four genes, and four have five genes; therefore, RNA 2 has a flexible genomic architecture and is apparently the most frequent genomic organization in the RNA 2 of varicosaviruses that includes four genes (21 members) or three genes (18 members).

The consensus gene junction sequences of the bisegmented varicosaviruses were determined to be 3′ AU(N)_5_UUUUUGCUCU 5′ (Table 2), while the gene junction sequences of all but one of the unsegmented varicosaviruses differed slightly in the 3´ end, being GU(N)_5_ instead of AU(N)_5_ (Table 2). Strikingly, the consensus gene junction of the unsegmented Torreya virus 1 (TorV1) was similar to that of the bisegmented varicosaviruses. The potential implication of this difference in the gene junctions needs to be explored since it could be linked to the basal evolutionary grouping of TorV1 (see below).

There is a great dearth of data on the potential functions of putative proteins, other than N and L, encoded by varicosaviruses, and, intriguingly, there were no conserved domains identified in these proteins. We grasped some shared identities, primarily for the cognate P3 (but also for several P2 proteins) (Table 1), though for most of the encoded proteins, the BlastP results were orphans, with no known signals or domains present and no clues towards their putative (or conserved) function. Thus, further studies should be focused on the functional characterization of these proteins to gain essential knowledge regarding the elusive proteome of varicosaviruses beyond the N and L proteins.

The pairwise aa sequence identities between the L proteins of all the reported varicosaviruses, including those identified in this study, showed great diversity and an overall low identity between the different varicosaviruses (Figure 2, Appendix A). Relatively low sequence identity is a common feature among rhabdovirus taxa, characterized by a high level of diversity in both the genome sequence and organization [10]. In addition, the overall low sequence identity among the novel viruses detected here and with the previously described varicosaviruses suggests that despite the many viruses identified in this study, there likely remains a significant amount of virus “dark matter” for yet-to-be-discovered varicosaviruses.

When we analyzed the diversity between the variants of viruses which are likely members of the same species, we found that proteins encoded by the Brassica virus 2, Spinach virus 1, and Sciadopitys virus 1 variants were very similar. On the other hand, proteins encoded by the Brassica virus 1, Lolium virus 1, and Melilotus virus 1 variants were quite diverse, but, nevertheless, they showed aa identities for the N and L proteins exceeding 80%. Thus, we tentatively propose an aa sequence identity of 80% across the L gene as the threshold for species demarcation in the *Varicosavirus* genus, a taxonomic criterion which had previously not been fully defined [10]. This threshold is strongly supported by the comparison of the L protein aa sequence of 60 viruses (Figure 2, Appendix A). Based on this criterion, all 39 novel viruses with their complete coding region assembled in this study should be considered as belonging to novel *Varicosavirus* species, which would increase the number of members of the genus by more than an order of magnitude.

Bejerman et al. [9] tentatively reported the first unsegmented varicosavirus, Pinus flexilis virus 1 (PiFleV1), which was associated with the gymnosperm *Pinus flexilis*. In this study, we complemented that result by the discovery of eight additional unsegmented varicosaviruses which were exclusively associated with gymnosperms (Table 1), some of which are linked to the same genus *Pinus* and present a significant co-evolution of viruses and hosts. These results robustly support a clade of gymnosperm-associated varicosaviruses with a distinct genome architecture, requiring the rewriting of a previously proposed key feature and fundamental marker of varicosaviruses: their genomic bisegmented nature. It is tempting to speculate that the unsegmented genomic architecture may be linked to the adaptation to gymnosperm hosts and a shared ancient evolutionary history of these viruses and hosts.

Interestingly, in the BlastP analyses of N, P2, and P3 of the gymnosperm-associated viruses, most of them had, as a best hit to the cognate proteins encoded by the putative bisegmented ASaV2 (Table 1), a virus apparently hosted by a parasitic plant of spruce (*Picea*, *Pinacea*). Furthermore, unexpectedly, the best hit of the putative P5 protein encoded on ASaV2 RNA2 was a fragment of the PiFleV1 L protein, while the deduced L protein on ASaV2 RN1 was not a best hit with PiFleV1, but instead, with the non-gymnosperm-linked MelRoV1 hosted by the Orobanchaceae parasitic plant *Melampyrum roseum*. Thus, we suspected that ASaV2 was potentially misassembled from fragments belonging to two different viruses. Consequently, we re-analyzed the original SRA data used by Sidhartan et al. [18] and were able to assemble two distinct varicosavirus genomes: one bisegmented genome presumably linked to the parasitic plant and one unsegmented genome most likely linked to spruce, which would support our hypothesis. We believe that there are several reasons that led to the original ASaV2 description: (i) the atypical and unexpected existence at the time of an unsegmented varicosavirus; (ii) the presence of two varicosaviruses in the very same sequencing library, which may be the first tentative evidence in the literature of co-infection of two varicosaviruses; and (iii) the fact that the sequence reads corresponding to the L gene region of the unsegmented varicosavirus were low, which may have affected the assembling pipelines used by the authors. All in all, independently verifying unexpected re-analysed SRA data may lead to a clearer understanding of the genomic structure of the mined RNA virus genomes. Nevertheless, the inability to return to the original biological material to replicate, confirm, and validate the assembled viral genome sequences is a significant limitation of the data mining approach for virus discovery. Thus, researchers must be cautious when analysing SRA public data for virus discovery and understand the preliminary nature of its results.

The phylogenetic analysis based on the deduced L protein aa sequences placed all unsegmented varicosaviruses, except TorV1, into a distinct clade. Interestingly, TorV1 was placed in a clade that was basal to all varicosaviruses (Figure 1). This distinct phylogenetic branching and clustering of the unsegmented viruses suggests that they share a unique evolutionary history among varicosaviruses. Moreover, this may suggest that bisegmented varicosaviruses are evolutionarily younger than unsegmented ones. It may also mean that a genome split in varicosa-like viruses occurred after the radiation of gymnosperms and angiosperms. Bisegmented varicosaviruses did not cluster according to their genomic organization, nor did they cluster with the plant species associated with each virus (Figure 1). For example, brassica virus 1 and brassica virus 2 were placed in distinct clades, while two viruses associated with orchids (Ophius virus 1 and Caladenia virus 1) were placed in different clusters, and monocot-associated viruses were not all grouped together. On the other hand, all varicosaviruses associated with ferns and liverworts belonged to the same cluster, which was also shared with previously reported varicosaviruses from these plant types, while most of the grass-associated varicosaviruses were also clustered together (Figure 1).

We generated a tanglegram to compare the virus phylogram and plant host cladogram to further explore virus–host relationships (Figure 3). This analysis showed that the viruses of some clades clearly co-diverged with their hosts, including the gymnosperm-associated virus clade, the SpV1 and Silene virus 1 clade, the grass-associated virus clade, and the clade of fern and liverworts viruses, suggesting a shared host–virus evolution in those clades (Figure 3). However, the tanglegram topology also indicated that for most of the varicosaviruses, there was no apparent concordant evolutionary history with their plant hosts, similar to what was previously reported for invertebrate and vertebrate rhabdoviruses [60].

Several lines of evidence suggest that varicosaviruses may be vertically transmitted: (i) a close host–virus co-evolution in some clades may reflect species isolation and a lack of horizontal transmission, (ii) some viruses detected in this study were identified from seed transcriptomics databases, and (iii) an emerging characteristic of persistent, chronic infections of several plant viruses which are likely vertically transmitted are latent/asymptomatic infections, a characteristic which appears to be shared with varicosaviruses. Thus, further studies should be carried out to elucidate the transmission mode of varicosaviruses beyond the fungal-transmitted LBVaV [11]. It is worth mentioning that even with the availability of thousands of RNAseq libraries of fungi and arthropods, we failed to detect any evidence of varicosaviruses in those organisms, which could suggest that vectors of varicosaviruses are rare or non-existent.

Before the era of data-driven virus discovery, few viruses had been identified in gymnosperms [61,62,63,64]. However, when data mining was applied to publicly available transcriptomes, many novel viruses were identified in this large group of higher plants, highlighting the rich and diverse gymnosperm virosphere, which still is largely unexplored. A distinct clade of gymnosperm-associated viruses was recently identified within amalgaviruses [65], while we recently described two distinct caulimovirids and geminivirids linked to the gnetophyte *Welwitschia mirabilis* [66]. Eight unsegmented varicosaviruses associated with gymnosperms were identified in this study, and another was discovered by Bejerman et al. [9]. Taken together, all of these recently discovered viruses in gymnosperms strongly suggest that they may have evolutionary trajectories that are distinct from those infecting angiosperms. Thus, it is likely that further exploration of additional gymnosperm datasets or new transcriptome studies of other gymnosperms will yield plenty of novel viruses with unique features, highlighting their close evolution with their hosts. The clear association between gymnosperm-associated viruses and their hosts likely indicates a close coevolution, which suggest an early adaptation of this group of viruses to infect gymnosperms. This hypothesis is also supported by the distinct genomic architecture and divergent evolutionary history among varicosaviruses, as shown in the phylogenetic tree, which are characterized by long branches and distinctive clustering. Taken together, the gymnosperm-associated varicosaviruses could be taxonomically classified in a novel genus within the family *Rhabdoviridae*, subfamily *Betarhabdovirinae,* for which we suggest the name “*Gymnorhavirus*”.

In summary, this study highlights the importance of the analysis of SRA public data as a valuable tool not only to accelerate the discovery of novel viruses, but also to gain insight into their evolution and to refine virus taxonomy. Using this approach, we looked for hidden varicosa-like virus sequences to unlock the veiled diversity of a largely neglected plant rhabdovirus genus, the varicosaviruses. Our findings, including an approximately 3.5-fold expansion of the current genomic diversity within the genus, resulted in the most complete phylogeny of varicosaviruses to date, and they shed new light on the genomic architecture, phylogenetic relationships, and evolutionary landscape of this unique group of plant rhabdoviruses. Future studies should assess many intriguing aspects of the biology and ecology of these viruses such as potential symptoms, vertical transmission, and putative vectors.

## Figures and Tables

**Figure 1 pathogens-11-01127-f001:**
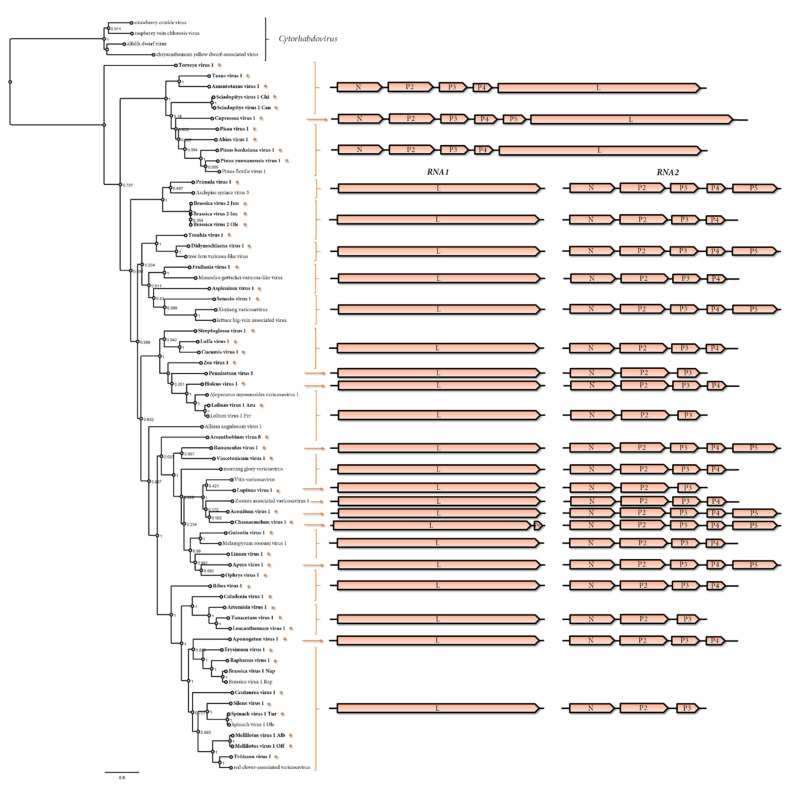
Left: Maximum-likelihood phylogenetic tree based on the amino acid sequence alignments of the complete L gene of all the varicosaviruses reported thus far and in this study. The scale bar indicates the number of substitutions per site. The node labels indicate fast tree support values. Four cytorhabdoviruses were used as outgroups. Right: Genomic organization of the varicosavirus sequences used in the phylogeny. An asterisk and bold font indicate those viruses identified in this study. The accession numbers of all the viruses are listed in Appendix A and Table 1.

**Figure 2 pathogens-11-01127-f002:**
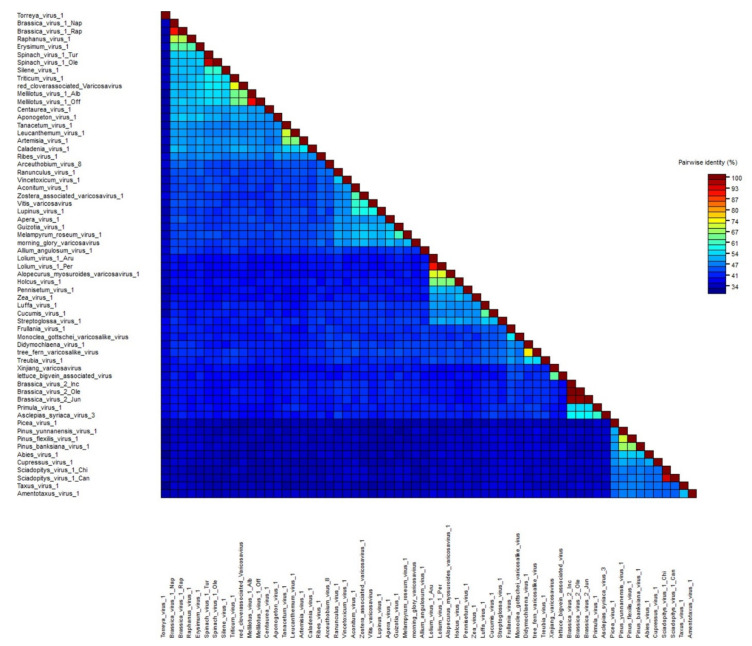
Pairwise identity matrix of the amino acid sequences of the varicosavirus complete L gene open reading frame generated using SDT v1.2 software [59]. GenBank accession numbers are listed in Appendix A and Table 1.

**Figure 3 pathogens-11-01127-f003:**
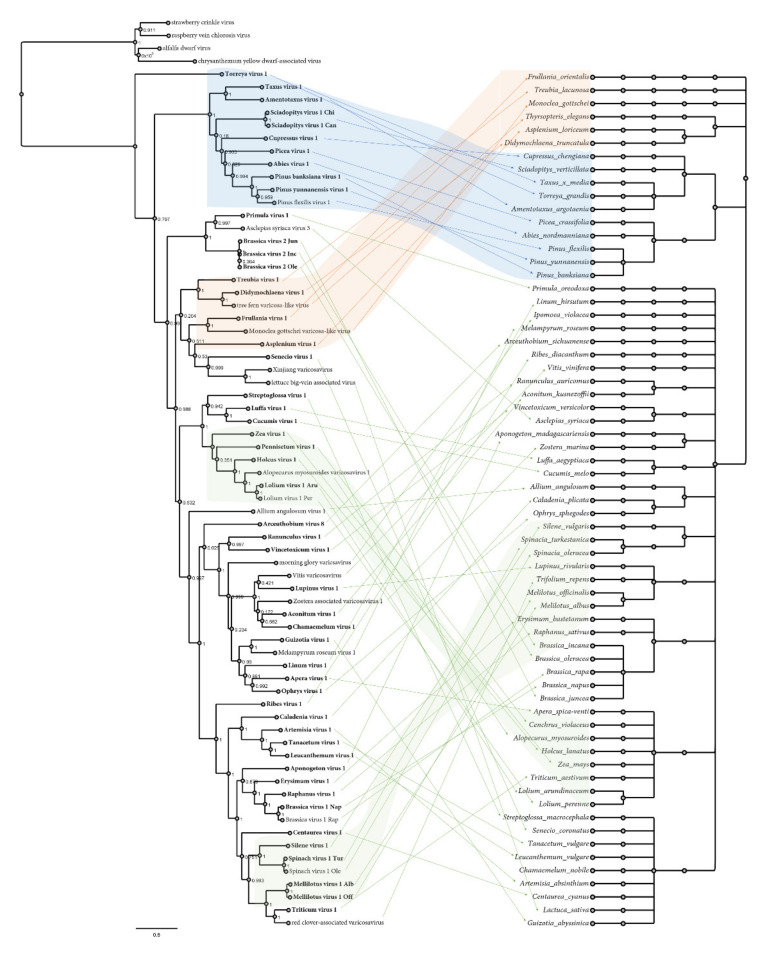
Tanglegram showing the phylogenetic relationships of the varicosaviruses (left), which are linked with the associated plant host(s) shown on the right. Links of well-supported clades of viruses to taxonomically related plant species are indicated in blue, orange, and green. A maximum likelihood phylogenetic tree of rhabdoviruses was constructed based on the conserved amino acid sequence of the complete L protein. Plant host cladograms were generated in phyloT v.2 based on NCBI taxonomy. Internal nodes represent the taxonomic structure of the NCBI taxonomy database, including species, genus, family, order, subclass, and sub-kingdom. Viruses identified in the present study are shown in bold font. The scale bar indicates the number of substitutions per site.

**Table 1 pathogens-11-01127-t001:** Summary of the novel varicosaviruses identified from the plant RNA-seq data available in the NCBI database. The acronyms of the best hits are listed in Appendix A.

Plant Host	Taxa/Family	Virus Name/Abbreviation	Bioproject ID/ Data Citation	Segment/Coverage	Length (nt)	Accession Number	Protein ID	Length (aa)	Highest Scoring Virus-Protein/E-Value/Query Coverage%/Identity% (Blast P)
Trojan fir (*Abies nordmannia*)	Gymnosperm/*Pinaceae*	Abies virus 1/ AbiV1	PRJNA387306/ University of Connecticut, USA	RNA1/30.97X	11,287	BK061731	N 2 3 4 L	430 420 317 163 2050	PiFleV1-N/9e-130/87/50.79 PiFleV1-P2/2e-18/57/28.05 PiFleV1-P3/2e-103/97/47.44 no hits PiFleV1-L/0.0/98/52.68
*Dwarf mistletoe* (*Arceuthobium sichuanense*)	dicot/ *Santalaceae*	Arceuthobium virus 8/ ArcV8	PRJNA307530/ [23]	RNA1/9.31X RNA2/72.35X	6628 4149	BK061732 BK061733	L N 2 3	2013 369 453 159	ASaV2-L/0.0/98/100 ZaVV1-N/1e-34/91/28.36 no hits no hits
Bei Wu Tou (*Aconitum kusnezoffii*)	dicot/ *Ranunculaceae*	Aconitum virus 1/ AcoV1	PRJNA670255/ [24]	RNA1/10.16X RNA2/105.03X	6483 5561	BK061734 BK061735	L N 2 3 4 5	2000 424 329 311 204 297	ZaVV1-L/0.0-97/61.18 ZaVV1-N/2e-115/99/43.82 VVV-P2/4e-36/80/32.13 ZaVV1-P3/5e-105/85/54.51 VVV-P4/1e-27/87/33.33 VVV-P5/5e-17/92/26.18
Catkin yew (*Amentotaxus argotaenia*)	Gymnosperm/ *Cephalotaxeae*	Amentotaxus virus 1/ AmeV1	PRJNA498605/ [25]	RNA1/109.96X	10,965	BK061736	N 2 3 4 L	391 431 314 187 2062	ASaV2-N/3e-111/94/45.95 PiFleV1-P2/1e-06/55/26.98 ASaV2-P3/4e-83/94/43.42 no hits PiFleV1-L/0.0/99/46.16
Common windgrass (*Apera spica-venti*)	monocot/ *Poaceae*	Apera virus 1/ ApeV1	PRJNA356380/ [26]	RNA1/11.98X RNA2/110.50X	6516 6552	BK061737 BK061738	L N 2 3 4 5	2027 447 363 298 196 444	MelRoV1-L/0.0/98/52.12 MelRoV1-N/2e-69/82/34.57 MelRoV1-P2/4e-17/75/26.37 MelRoV1-P3/2e-80/97/41.25 no hits no hits
Lace plant (*Aponogeton madagascariensis*)	monocot/ *Aponogetonaceae*	Aponogeton virus 1/ ApoV1	PRJNA591467/ [27]	RNA1/36.42X RNA2/81.25X	6678 5628	BK061739 BK061740	L N 2 3 4	2022 435 454 300 174	BrRV1-L/0.0/98/52.7 BrRV1-N/7e-81/88/37 no hits TfVV-P3/2e-45/96/34 BrRV1-P3/0.003/73/25
Wormwood (*Artemisia absinthium*)	dicot/ *Asteraceae*	Artemisia virus 1/ ArtV1	PRJNA371565/ [28]	RNA1/33.06X RNA2/50.30X	7373 4497	BK061741 BK061742	L N 2 3	2020 453 494 174	BrRV1-L/0.0/98/49.18 BrRV1-N/3e-45/76/28.90 no hits no hits
Common milkweed (*Asclepias syriaca*)	dicot/ *Apocynaceae*	*Asclepias syriaca* virus 3 AscSyV3	PRJNA210776/ [29]	RNA1/37.86X RNA2/138.94X	6506 6280	BK061743 BK061744	L N 2 3 4 5	2021 453 370 286 160 393	TfVV-L/0.0/94/42.62 TfVV-N/3e-39/78/32.13 no hits TfVV-P3/73-32/78/29.26 no hits no hits
Beautiful tree fern (*Asplenium loriceum*)	*Polypodiophyta/* *Aspleniaceae*	Asplenium virus 1/ AspV1	PRJNA281136/ [30]	RNA1/4.51X RNA2/8.91X	6287 * 4371 *	BK061745 BK061746	L N 2 3 4	1957 * 396 490 294 127 *	TfVV-L/0.0/98/43.81 TfVV-N/2e-79/90/37.82 no hits TfVV-P3/1e-45/87/33.33 no hits
Shortpod mustard (*Brassica incana*)¡	dicot/ *Brassicaceae*	Brassica virus 2_Inc/ BrV2_Inc	PRJNA428769/ [31]	RNA1/11.89X RNA2/14.63X	6316 5616	BK061747 BK061748	L N 2 3 4	2032 591 459 282 141	TfVV-L/0.0/99/41.86 LoPV1-N/1e-31/58/27.93 no hits TfVV-P3/9e-33/91/29.32 no hits
Indian mustard (*Brassica juncea* var. rugosa)	dicot/ *Brassicaceae*	Brassica virus 2_Jun/ BrV2_Jun	PRJNA290942/ [32]	RNA1/80.91X RNA2/950.63X	6316 5537	BK061749 BK061750	L N 2 3 4	2032 591 459 282 141	TfVV-L/0.0/99/41.57 LoPV1-N/6e-31/58/27.65 no hits TfVV-P3/1e-32/91/29.32 no hits
Chinese kale (*Brassica oleracea* var. alboglabra)	dicot/ *Brassicaceae*	Brassica virus 2_Ole/ BrV2_Ole	PRJNA525713/ [33]	RNA1/11.03X RNA2/66.34X	6316 5647	BK061751 BK061752	L N 2 3 4	2032 591 459 282 141	TfVV-L/0.0/99/41.81 LoPV1-N/7e-32/58/27.93 no hits TfVV-P3/8e-33/91/29.32 no hits
Crab-lipped spider orchid (*Caladenia plicata*)	monocot/ *Orchidaceae*	Caladenia virus 1/ CalV1	PRJNA384875/ [34]	RNA1/10.51X RNA2/52.44X	6454 5011	BK061755 BK061756	L N 2 3 4	2024 449 468 293 165	BrRV1-L/0.0/98/50.17 BrRV1-N/1e-64/97/32.43 no hits TfVV-P3/1e-43/86/34.78 BrRV1-P3/3e-07/61/31.13
Conrflower (*Centaurea cyanus*)	dicot/ *Asteraceae*	Centaurea virus 1/ CenV1	PRJNA371565/ [28]	RNA1/63.11X RNA2/159.93X	6789 4567	BK061757 BK061758	L N 2 3	2019 469 501 111	BrRV1-L/0.0/98/50.50 BrRV1-N/6e-48/73/30.72 no hits no hits
Chamomile (*Chamaemelum nobile*)	dicot/ *Asteraceae*	Chamaemelum virus 1/ ChaV1	PRJNA382469/ [35]	RNA1/21.33X RNA2/234.84X	6670 * 5957	BK061759 BK061760	L P6 N 2 3 4 5	1916 * 171 426 346 305 255 330	VVV-L/0.0/99/58.85 no hits ZaVV1-N/2e-105/95/41.40 VVV-P2/2e-19/84/30.28 VVV-P3/5e-97/94/49.14 ZaVV1-P4/3e-05/70/22.1 VVV-P5/3e-22/85/29.14
Melon (*Cucumis melo*)	dicot/ *Cucurbitaceae*	Cucumis virus 1/ CucV1	PRJNA381300/ [36]	RNA1/47.79X RNA2/60.05X	6919 5322	BK061761 BK061762	L N 2 3 4	2034 341 404 285 119	AMVV1-L/0.0/99/47.47 InPRV-N/4e-77/98/38.71 no hits TfVV-P3/1e-46/91/34.21 no hits
Chen cypress (*Cupressus chengiana*)	*Gymnosperm/* *Cupressaceae*	Cupressus virus 1/ CupV1	PRJNA556937/ [37]	RNA1/32.13X	12143	BK061763	N 2 3 4 5 L	379 447 313 187 168 2055	ASaV2-N/2e-106/97/44.59 ASaV2-P2/5e-30/67/30.86 ASaV2-P3/2e-100/84/53.38 no hits no hits PiFleV1-L/0.0/99/48.68
Tree maidenhair fern (*Didymochlaena truncatula*)	*Polypodiophyta/* *Hypodeatiaceae*	Didymochlaena virus 1/ DidV1	PRJNA422112/ [38]	RNA1/8.88X RNA2/52.28X	6319 5924	BK061764 BK061765	L N 2 3 4 5	2044 386 394 292 187 374	TfVV-L/0.0/100/74.17 TfVV-N/0.0/100/72.75 TfVV-P2/7e-74/96/40.26 TfVV-P3/2e-159/99/70.69 TfVV-P4/5e-23/88/30.72 TfVV-P5/0.0/97/64.11
Wallflower (*Erysimum bastetanum*)	dicot/ *Brassicaceae*	Erysimum virus 1/ EryV1	PRJNA607615/ [39]	RNA1/271.24X RNA2/516.22X	6676 3980	BK061766 BK061767	L N 2 3	1985 439 404 172	BrRV1-L/0.0/99/62-34 BrRV1-N/3e-90/99/33.86 no hits BrRV1-P3/4e-26/100/31.4
Liverwort (*Frullania orientalis*)	*Marchantiophyta/* *Frullaniaceae*	Frullania virus 1/ FruV1	PRJNA505755/ Fairylake Botanical Garden, China	RNA1/11.60X RNA2/8.20X	6458 4363	BK061768 BK061769	L N 2 3 4	2033 372 336 289 148	MgVV-L/0.0/98/54.77 MgVV-N/2e-94/97/43.96 MgVV-P2/8e-05/56/27.27 MgVV-P3/5e-85/89/47.49 MgVV-P4/4e-05/70/29.81
Noug (*Guizotia abyssinica*)	dicot/ *Asteraceae*	Guizotia virus 1/ GuiV1	PRJNA371565/ [28]	RNA1/153.49X RNA2/1192.66X	6457 4722	BK061770 BK061771	L N 2 3 4	2007 434 340 262 307	MelRoV1-L/0.0/98/60.42 MelRoV1-N/3e-103/82/43.96 MelRoV1-P2/7e-22/85/24.53 no hits no hits
Common velvet grass (*Holcus lanatus*)	monocot/ *Poaceae*	Holcus virus 1/ HolV1	PRJEB11654/ [40]	RNA1/19.48X RNA2/29.44X	6571 4397	BK061772 BK061773	L N 2 3 4	2031 476 286 211 161	AMVV1-L/0.0/98/65.12 LoPV1-N/8e-132/77/51.23 LoPV1-P2/5e-23/56/33.33 LoPV1-P2/8e-12/63/29.76 LoPV1-P3/1e-49/90/51.72
Oxeye daisy (*Leucanthemum vulgare*)	dicot/ *Asteraceae*	Leucanthemum virus 1/ LeuV1	PRJNA371565/ [28]	RNA1/141.76X RNA2/229.85X	6763 4775	BK061774 BK061775	L N 2 3	2021 448 520 167	BrRV1-L/0.0/98/49.63 BrRV1-N/3e-42/71/32.11 no hits no hits
Downy flax (*Linum hirsutum*)	dicot/ *Linaceae*	Linum virus 1/ LinV1	PRJEB21674/ 1000 Plant (1KP) Transcriptomes Initiative	RNA1/26.47X RNA2/119.90X	5999 * 6330	BK061776 BK061777	L N 2 3 4	1940 * 450 463 313 260	MelRoV1-L/0.0/94/53.78 MelRoV1-/3e-69/82/33.96 no hits MelRoV1-P3/7e-81/88/42.39 no hits
Sponge gourd (*Luffa aegyptiaca*)	dicot/ *Cucurbitaceae*	Luffa virus 1/ LufV1	PRJNA390566/ Mylne, J., The University of Western Australia	RNA1/16.47X RNA2/11.32X	6693 4961	BK061780 BK061781	L N 2 3 4	2032 487 366 286 126	LoPV1-L/0.0/99/49.04 InPRV-N/7e-84/86/36.93 no hits TfVV-P3/3e-53/81/41.7 no hits
Riverbank lupine (*Lupinus rivularis*)	dicot/ *Fabaceae*	Lupinus virus 1/ LupV1	PRJNA318864/ [41]	RNA1/14.64X RNA2/97.57X	6688 4042 *	BK061782 BK061783	L N 2 3	1997 426 497 116 *	ZaVV1-L/0.0/99/56.91 ZaVV1-N/2e-83/99/36.92 ZaVV1-P2/3e-14/39/28.99 no hits
Sweet clover (*Melilotus spp*)	dicot/ *Fabaceae*	Melilotus virus 1_Alb/ MelV1_Alb	PRJNA647665/ [42]	RNA1/30.69X RNA2/98.21X	6657 3985	BK061784 BK061785	L N 2 3	2019 430 393 189	RCaVV-L/0.0/99/64.97 RCaVV-N/5e-80/93/33.5 RCaVV-P2/0.001/42/27.54 RCaVV-P3/8e-25/88/35.12
Sweet clover (*Melilotus spp*)	dicot/ *Fabaceae*	Melilotus virus 1_Off/ MelV1_Off	PRJNA751393/ [43]	RNA1/12.15X RNA2/25.36X	6433 3781	BK061786 BK061787	L N 2 3	2019 430 399 191	RCaVV-L/0.0/99/65.37 RCaVV-N/5e-77/91/33.33 RCaVV-P2/0.002/42/28.14 RCaVV-P3/5e-23/87/34.52
Early spider orchid (*Ophrys sphegodes*)	monocot/ *Orchidaceae*	Ophrys virus 1/ OphV1	PRJNA574279/ [44]	RNA1/7.72X RNA2/206.15X	6134 * 5036	BK061788 BK061789	L N 2 3 4	1988 * 447 466 293 214	MelRoV1-L/0.0/99/56.95 MelRoV1-N/4e-97/96/37.1 MelRoV1-P2/4e-23/54/28.9 MelRoV1-P3/2e-84/91/43.87 MelRoV1-P4/0.009/63/26.39
Purple Grass (*Pennisetum violaceum*)	monocot/ *Poaceae*	Pennisetum virus 1/ PenV1	PRJNA282366/ Suja George, M.S Swaminathan Research Foundation, India	RNA1/44.59X RNA2/112.25X	6284 3407	BK061790 BK061791	L N 2 3	2033 451 286 151	LoPV1-L/0.0/98/51.27 LoPV1-N/5e-79/75/40.52 no hits LoPV1-P3/4e-12/83/30.16
Qinghai spruce (*Picea crassifolia*)	Gymosperm/ *Pinaceae*	Picea virus 1/ PicV1	PRJNA307530/ [23]	RNA1/5.86X	11,193	BK061792	N 2 3 4 L	382 452 318 174 2051	ASaV2-N/0.0/100/100 ASaV2-P2/0.0/100/100 ASaV2-P3/0.0/100/100 ASaV2-P4/0.0/100/100 PiFleV1-L/0.0/99/49.12
Jack pine (*Pinus banksiana*)	Gymosperm/ *Pinaceae*	*Pinus banksiana* virus 1/ PiBanV1	PRJNA524866/ [45]	RNA1/97.66X	11276	BK061793	N 2 3 4 L	406 433 317 175 2048	PiFleV1-N/0.0/100/68.72 PiFleV1-P2/3e-48/57/39.2 PiFleV1-P3/1e-161/100/64.78 PiFleV1-P4/3e-17/65/36.84 PiFleV1-L/0.0/99/65.35
Yunnan pine (*Pinus yunnanensis*)	Gymosperm/ *Pinaceae*	*Pinus yunnanensis* virus 1/PiYunV1	PRJNA507489/ [46]	RNA1/36.47X	12,057	BK061794	N 2 3 4 L	411 440 319 204 2048	PiFleV1-N/0.0/93/70.5 PiFleV1-P2/7e-48/97/35.49 PiFleV1-P3/8e-145/100/62.38 PiFleV1-P4/7e-30/75/38.46 PiFleV1-L/0.0/98/70.33
Spendlor primrose (*Primula oreodoxa*)	dicot/ *Primulaceae*	Primula virus 1/ PriV1	PRJNA544868/ [47]	RNA1/7.72X RNA2/149.23X	6352 6283	BK061795 BK061796	L N 2 3 4 5	2022 435 352 288 145 384	TfVV-L/0.0/98/42.3 TfVV-N/1e-40/74/33.33 no hits TfVV-P3/2e-28/75/29.55 no hits no hits
Goldilocks buttercup (*Ranunculus auricomus*)	dicot/ *Ranunculaceae*	Ranunculus virus 1/ RanV1	PRJNA217403/ [48]	RNA1/29.64X RNA2/163.27X	6481 6269	BK061797 BK061798	L N 2 3 4 5	2034 529 438 307 200 330	MelRoV1-L/0.0/98/49.85 MelRoV1-N/2e-65/63/34.63 MelRoV1.P2/4e-08/26/27.83 ZaVV1-P3/2e-59/79/42.86 no hits no hits
Radish (*Raphanus sativus*)	dicot/ *Brassicaceae*	Raphanus virus 1/ RapV1	PRJNA539856/ [49]	RNA1/165.02X RNA2/521.73X	6410 4144	BK061799 BK061800	L N 2 3	2016 439 411 175	BrRV1-L0.0/99/68.31 BrRV1-N/1e-135/100/46.94 BrRV1-P2/5e-14/61/28.57 BrRV1-P3/6e-34/98/37.5
Siberian currant (*Ribes diacanthum*)	dicot/ *Grossulariaceae*	Ribes virus 1/ RibV1	PRJNA407394/ [50]	RNA1/6.29X RNA2/33.97X	6323 5201	BK061801 BK061802	L N 2 3 4	2017 372 402 301 194	SpV1-L/0.0/98/47.29 TfVV-N/1e-60/90/36.01no hits TfVV-P3/2e-45/82/33.33 no hits
Japanese umbrella pine (*Sciadopitys verticillata*)	Gymnosperm/ *Sciadopityaceae*	Sciadopitys virus 1_Chi/ SciV1_Chi	PRJNA396655/ Institute of Botany, CAS, China	RNA1/98.99X	11,224	BK061803	N 2 3 4 L	389 466 315 168 2054	ASaV2-N/1e-111/95/43.13 ASaV2-P2/1e-22/60/30.14 ASaV2-P3/4e-104/95/48.23 PiFleV1-P4/3e-05/67/26.32 PiFleV1-L/0.0/99/46.13
Japanese umbrella pine (*Sciadopitys verticillata*)	Gymnosperm/ *Sciadopityaceae*	Sciadopitys virus 1_Can/ SciV1_Can	PRJEB4921/ [51]	RNA1/14.02X	11,132	BK061804	N 2 3 4 L	389 466 314 168 2071	ASaV2-N/1e-111/95/43.67 ASaV2-P2/8e-22/60/29.87 ASaV2-P3/2e-105/95/48.23 PiFleV1-P4/7e-07/80/25.93 PiFleV1-L/0.0/99/45.88
Wooly grassland senecio (*Senecio coronatus*)	dicot/ *Asteraceae*	Senecio virus 1/ SenV1	PRJNA312157/ [52]	RNA1/10.59X RNA2/93.61X	6173 * 5617	BK061805 BK061806	L N 2 3 4 5	2031 * 376 345 294 147 370	LBVaV-L/0.0/98/42.8 PhPV1/2e-132/98/51.98 no hits PhPV1-P3/9e-124/87/56.64 no hits XVV-L/2e-08/29/30
Bladder campion (*Silene vulgaris*)	dicot/ *Caryophyllaceae*	Silene virus 1/ SilV1	PRJNA104951/ [53]	RNA1/29.59X RNA2/77.05X	6391 4363	BK061807 BK061808	L N 2 3	2019 445 509 179	SpV1-0.0/99/59.91 SpV1-N/4e-65/91/33.99 SpV1-P2/2e-13/61/24.07 BrRV1-P3/0.001/97/24.29
Broadhead daisy (*Streptoglossa macrocephala*)	dicot/ *Asteraceae*	Streptoglossa virus 1/ StrV1	PRJNA371565/ [28]	RNA1/131.33X RNA2/140.03X	6776 5130	BK061813 BK061814	L N 2 3 4	2023 449 333 287 162	LoPV1-L/0.0/99/49.09 InPRV-N3e-86/99/36.01 no hits PhPV1-P3/2e-43/91/32.2 no hits
Tansy (*Tanacetum vulgare*)	dicot/ *Asteraceae*	Tanacetum virus 1/ TanV1	PRJNA646340/ [54]	RNA1/10.19X RNA2/239.11X	6888 4608	BK061815 BK061816	L N 2 3	2020 447 505 176	BrRV1-L/0.0/98/49.03 BrRV1-L/8e-52/88/30.56 no hits RCaVV-P3/3e-05/73/30.60
Hybrid yew (*Taxus media*)	Gymnosperm/ *Taxaceae*	Taxus virus 1/ TaxV1	PRJNA497542/ [55]	RNA1/57.28X	11,174	BK061817	N 2 3 4 L	382 417 310 201 2057	ASaV2-N/7e-111/96/43.55 ASaV2-P2/1e-18/68/26.28 ASaV2-P3/3e-94/93/45.25 no hits PiFleV1-L/0.0/98/46.81
Chinese nutmeg yew (*Torreya grandis*)	Gymnosperm/ *Taxaceae*	Torreya virus 1/ TorV1	PRJNA498605 [25]	RNA1/59.04X	10,253	BK061818	N 2 3 4 L	379 339 283 152 2002	TfVV-N/2e-57/93/32.5 no hits TfVV-P3/4e-28-67/36.27 no hits TfVV-L/0.0/97/35.4
Liverwort (*Treubia lacunosa*)	*Marchantiophyta/* *Treubiaceae*	Treubia virus 1/ TreV1	PRJNA505755/ Fairylake Botanical Garden, China	RNA1/364.20X RNA2/350.53X	6684 4940	BK061819 BK061820	L N 2 3 4	2040 392 395 288 153	TfVV-L/0.0/99/54.2 TfVV-N/3e-116/99/46.27 TfVV-P2/0.015/56/24.34 TfVV-P31e-114/85/55.07 no hits
Wheat (*Triticum aestivum*)	monocot/ *Poaceae*	Triticum virus 1/ TriV1	PRJNA558380/ [56]	RNA1/10.25X RNA2/16.64X	6290 4103	BK061821 BK061822	L N 2 3	2019 430 451 179	RCaVV-L/0.0/99/72.58 RCaVV-N/8e-135/99/46.26 RCaVV-P2/2e-32/67/30.70 RCaVV-P3/1e-48/100/44.13
Variegated swallow-wort (*Vincetoxicum versicolor*)	dicot/ *Apocynaceae*	Vincetoxicum virus 1/ VinV1	PRJNA599262/ [57]	RNA1/56.05X RNA2/140.76X	6598 4655	BK061823 BK061824	L N 2 3 4	2037 430 356 307 174	MelRoV1-L/0.0/99/48.19 ZaVV1-N/7e-63/76/35 MelRoV1-P2/2e-08/68/21.15 MelRoV1-P3/63-51/80/36.44 no hits
Corn (*Zea mays*)	monocot/ *Poaceae*	Zea virus 1/ ZeaV1	PRJNA407369/ [58]	RNA1/6.25X RNA2/40.88X	6345 4607	BK061825 BK061826	L N 2 3 4	2037 483 353 286 158	AMVV1-L/0.0/99/49.07 AMVV1-N/2e-90/76/40.92 LoPV1-P2/4e-08/63/24.89 TfVV-P3/6e-48/94/31.11 LoPV1-P3/1e-09/86/29.2

* partial sequence.

**Table 2 pathogens-11-01127-t002:** Consensus varicosavirus gene junction sequences.

Virus *	3′ end mRNA	Intergenic Spacer	5′ end mRNA
AbiV1	CU(N)_5_UUUUU	G	CUCU
ArcV8	AU(N)_5_UUUUU	G	CUCU
AcoV1	AU(N)_5_UUUUU	G	CUCU
AmeV1	CU(N)_5_UUUUU	G	CUCU
ApeV1	AU(N)_5_UUUUU	G	CUCU
ApoV1	AU(N)_5_UUUUU	G	CUCU
ArtV1	AU(N)_5_UUUUU	G	CUCU
AscSyV3	AU(N)_5_UUUUU	G	CUCU
AspV1	AU(N)_5_UUUUU	G	CUCU
BrV2	AU(N)_5_UUUUU	G	CUCU
CalV1	AU(N)_5_UUUUU	G	CUCU
CenV1	AU(N)_5_UUUUU	G	CUCU
ChaV1	AU(N)_5_UUUUU	G	CUCU
CucV1	AU(N)_5_UUUUU	G	CUCU
CupV1	CU(N)_5_UUUUU	G	CUCU
DidV1	AU(N)_5_UUUUU	G	CUCU
EryV1	AU(N)_5_UUUUU	G	CUCU
FruV1	AU(N)_5_UUUUU	G	CUCU
GuiV1	AU(N)_5_UUUUU	G	CUCU
HolV1	AU(N)_5_UUUUU	G	CUCU
LeuV1	AU(N)_5_UUUUU	G	CUCU
LinV1	AU(N)_5_UUUUU	G	CUCU
LufV1	AU(N)_5_UUUUU	G	CUCU
LupV1	AU(N)_5_UUUUU	G	CUCU
MelV1	AU(N)_5_UUUUU	G	CUCU
OphV1	AU(N)_5_UUUUU	G	CUCU
PenV1	AU(N)_5_UUUUU	G	CUCU
PicV1	CU(N)_5_UUUUU	G	CUCU
PiBanV1	CU(N)_5_UUUUU	G	CUCU
PiYunV1	CU(N)_5_UUUUU	G	CUCU
PriV1	AU(N)_5_UUUUU	G	CUCU
RanV1	AU(N)_5_UUUUU	G	CUCU
RapV1	AU(N)_5_UUUUU	G	CUCU
RibV1	AU(N)_5_UUUUU	G	CUCU
SciV1	CU(N)_5_UUUUU	G	CUCU
SenV1	AU(N)_5_UUUUU	G	CUCU
SilV1	AU(N)_5_UUUUU	G	CUCU
StrV1	AU(N)_5_UUUUU	G	CUCU
TanV1	AU(N)_5_UUUUU	G	CUCU
TaxV1	CU(N)_5_UUUUU	G	CUCU
TorV1	AU(N)_5_UUUUU	G	CUCU
TreV1	AU(N)_5_UUUUU	G	CUCU
TriV1	AU(N)_5_UUUUU	G	CUCU
VinV1	AU(N)_5_UUUUU	G	CUCU
ZeaV1	AU(N)_5_UUUUU	G	CUCU
AAnV1	AU(N)_5_UUUUU	G	CUCU
AMVV1	AU(N)_5_UUUUU	G	CUCU
BrV1	AU(N)_5_UUUUU	G	CUCA
LBVaV	AU(N)_5_UUUUU	G	CUCU
LoV1	AU(N)_5_UUUUU	G	CUCU
MelRoV1	AU(N)_5_UUUUU	G	CUCU
MGVV	AU(N)_5_UUUUU	G	CUCU
MgVV	AU(N)_5_UUUUU	G	CUCU
PhPiV1	AU(N)_5_UUUUU	G	CUCU
PiFleV1	GU(N)_5_UUUUU	G	CUCU
RCaVV	AU(N)_5_UUUUU	G	CUCU
SpV1	AU(N)_5_UUUUU	G	CUCU
TfVV	AU(N)_5_UUUUU	G	CUCU
VVV	AU(N)_5_UUUUU	G	CUCU
XVV	AU(N)_5_UUUUU	G	CUCU
ZaVV1	AU(N)_5_UUUUU	G	CUCU

The consensus gene junction sequences of the viruses identified in this study are highlighted in light grey. * Names and abbreviations of newly identified viruses are listed in Table 1; while the names and abbreviations of known viruses are listed in Appendix A.

## Data Availability

The nucleotide sequence data reported are available in the Third Party Annotation Section of the DDBJ/ENA/GenBank databases under the accession numbers TPA: BK061731-BK061826. These sequences are available as Appendix A.

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
