# Peer review of "Unlocking the Hidden Genetic Diversity of Varicosaviruses, the Neglected Plant Rhabdoviruses"

_pathogens, 2022, doi:10.3390/pathogens11101127_

Round 1

Reviewer 1 Report

The manucript of Bejerman et al. describes the determination of 45 complete or nearly complete varicosavirus genomes by a data mining approach and their further analyses, extending considerably the view on the varicosavirus diversity.

The methodological and bioinformatics workflow is described properly and the obtained results are of interest.

Minor suggestions/remarks:

Line 113, Table 1 needs to be better formatted, as the lines in last column (Highest scoring virus ...) do not fit with previous ones

Line 113, Table 1: throughout the novel varicosaviruses described in Table 1, there is 46 proteins with no hit obtained. Can authors comment/discuss it in their article? Is it a consequence of recombination with a yet undiscovered virus or another raison?

 Lines 156, 217, 219, 220: the use of the term “strain” is innapropriate here, I suggest rather the term “variant(s)” or “sequence variant(s)”

A general comment:

As seen in case of Arceuthobium sichuanense-associated virus 2, its genome was previously miss-assembled from fragments belonging to two different viruses. This show the difficulties to characterise a novel virus only based on in silico data. Although the authors made an important (and I believe that very powerful) bioinformatics effort, all the novel genomes were obtained only from in-silico data (which were not confirmed by RT-PCR and partial sequencing) and not biologically isolated. I understand that it was not possible in this work, but at least, authors can face this problem in the Discussion section (e.g. near the line 221). Can the data showing the main coverage depth be presented for novel genomes (in Supplementary), i.e. after remapping of HTS reads to the novel genomes.

Author Response

The manuscript of Bejerman et al. describes the determination of 45 complete or nearly complete varicosavirus genomes by a data mining approach and their further analyses, extending considerably the view on the varicosavirus diversity.

The methodological and bioinformatics workflow is described properly and the obtained results are of interest.

We thank reviewer #1 for taking the time to thoroughly assess our MS and provide suggestions which improved the MS.

General comments:

As seen in case of Arceuthobium sichuanense-associated virus 2, its genome was previously miss-assembled from fragments belonging to two different viruses. This show the difficulties to characterise a novel virus only based on in silico data. Although the authors made an important (and I believe that very powerful) bioinformatics effort, all the novel genomes were obtained only from in-silico data (which were not confirmed by RT-PCR and partial sequencing) and not biologically isolated. I understand that it was not possible in this work, but at least, authors can face this problem in the Discussion section (e.g. near the line 221).

We concur with the reviewer comment. We understand the limitations of our study and the prudence needed when suggesting conclusions emerging from in silico based works. A sentence that reads as: “Nevertheless, the inability to return to the original biological material to replicate, confirm and validate the assembled viral genome sequences is a significant limitation of the data mining approach for virus discovery. Thus, researchers need to be cautious when analysing SRA public data for virus discovery and understand the preliminary nature of its results.” was added in Lines 216-226.

Can the data showing the main coverage depth be presented for novel genomes (in Supplementary), i.e. after remapping of HTS reads to the novel genomes.

Thank you for this useful suggestion. We have added the main coverage depth values in a column in Table 1

Minor comments:

Line 113, Table 1 needs to be better formatted, as the lines in last column (Highest scoring virus ...) do not fit with previous ones

We have redesigned Table 1 merging some columns and adding a new one to improve the format and clarity of this Table.

Line 113, Table 1: throughout the novel varicosaviruses described in Table 1, there is 46 proteins with no hit obtained. Can authors comment/discuss it in their article? Is it a consequence of recombination with a yet undiscovered virus or another raison?

We have commented this point in Lines 166-172.

 Lines 156, 217, 219, 220: the use of the term “strain” is innapropriate here, I suggest rather the term “variant(s)” or “sequence variant(s)”

We replaced strain with variants. Thank you.

Reviewer 2 Report

The paper by Bejerman et al. reports on a discovery of 45 new varicosa- and varicosa-like (Rhabdoviridae) virus genomes in publicly available transcriptomics databases generated for a variety of gymnosperm, angiosperm, and fern species. It contains very interesting, novel and significant information enriching the virology literature and expanding our understanding of the genetic diversity and evolution of rhabdoviruses. In addition, the paper is well-written overall. Only a few issues need to be addressed, these are listed below:

1) Table 1 listing all the databases, where the virus genomes were assembled from, may need some corrections to make it easier to comprehend. Currently, all species are grouped alphabetically based on Latin names, which is fine. However, the same first column also has the host type in addition to the species name; this information is essentially lost for the reader. My recommendation is to add another column where the host type may be listed as a letter, say "G" for gymnosperm, "M" for monocots, "D" for dicots, etc. This will open up this important host type information which is discussed extensively later.

2) l. 230 - 'gymnosperm' should not be italicized

3) l. 262 - Actually, basal to all varicosaviruses. This may suggest that bisegmented varicosaviruses are evolutionarily younger than unsegmented ones. It may also mean that genome split in varicosa-like viruses occurred after the radiation of gymnosperms and angiosperms. 

4) ll. 303-304 - See a much older example here: Arch Virol. 2002 Nov;147(11):2207-14. doi: 10.1007/s00705-002-0876-5. It predated the HTS era. 

Author Response

The paper by Bejerman et al. reports on a discovery of 45 new varicosa- and varicosa-like (Rhabdoviridae) virus genomes in publicly available transcriptomics databases generated for a variety of gymnosperm, angiosperm, and fern species. It contains very interesting, novel and significant information enriching the virology literature and expanding our understanding of the genetic diversity and evolution of rhabdoviruses. In addition, the paper is well-written overall.

We thank reviewer #2 for taking the time to thoroughly assess our MS and provide suggestions which improved the MS.

Minor comments:

Table 1 listing all the databases, where the virus genomes were assembled from, may need some corrections to make it easier to comprehend. Currently, all species are grouped alphabetically based on Latin names, which is fine. However, the same first column also has the host type in addition to the species name; this information is essentially lost for the reader. My recommendation is to add another column where the host type may be listed as a letter, say "G" for gymnosperm, "M" for monocots, "D" for dicots, etc. This will open up this important host type information which is discussed extensively later.

We have redesigned Table 1 merging some columns and adding a new one to improve the format and clarity of this Table. We added a column named as Taxa/family to provide additional data on virus host.

L230 - 'gymnosperm' should not be italicized

corrected

L262 - Actually, basal to all varicosaviruses. This may suggest that bisegmented varicosaviruses are evolutionarily younger than unsegmented ones. It may also mean that genome split in varicosa-like viruses occurred after the radiation of gymnosperms and angiosperms. 

We thank the reviewer for this insightful suggestion. A sentence that reads “Moreover, this may suggest that bisegmented varicosaviruses are evolutionarily younger than unsegmented ones. It may also mean that genome split in varicosa-like viruses occurred after the radiation of gymnosperms and angiosperms” was added in Lines 231-233.

 L303-304 - See a much older example here: Arch Virol. 2002 Nov;147(11):2207-14. doi: 10.1007/s00705-002-0876-5. It predated the HTS era. 

This important reference was added to the manuscript as number 27.